# Longitudinal Analysis of Humoral and Cellular Immune Response up to 6 Months after SARS-CoV-2 BA.5/BF.7/XBB Breakthrough Infection and BA.5/BF.7-XBB Reinfection

**DOI:** 10.3390/vaccines12050464

**Published:** 2024-04-26

**Authors:** Xun Wang, Meng Zhang, Kaifeng Wei, Chen Li, Jinghui Yang, Shujun Jiang, Chaoyue Zhao, Xiaoyu Zhao, Rui Qiao, Yuchen Cui, Yanjia Chen, Jiayan Li, Guonan Cai, Changyi Liu, Jizhen Yu, Wenhong Zhang, Faren Xie, Pengfei Wang, Yanliang Zhang

**Affiliations:** 1Shanghai Pudong Hospital, Fudan University Pudong Medical Center, State Key Laboratory of Genetic Engineering, MOE Engineering Research Center of Gene Technology, School of Life Sciences, Shanghai Institute of Infectious Disease and Biosecurity, Fudan University, Shanghai 200437, China; wang_xunfudan@fudan.edu.cn (X.W.); lc876278963@163.com (C.L.); chaoyuejaul@163.com (C.Z.); xiaoyu_zhao@fudan.edu.cn (X.Z.); 21110700072@m.fudan.edu.cn (R.Q.); 21210700060@m.fudan.edu.cn (Y.C.); 22210700006@m.fudan.edu.cn (Y.C.); 22210700117@m.fudan.edu.cn (J.L.); caiguonan480@gmail.com (G.C.); 23210700136@m.fudan.edu.cn (C.L.); 23210700173@m.fudan.edu.cn (J.Y.); 2Department of Infectious Diseases, Nanjing Hospital of Chinese Medicine Affiliated to Nanjing University of Chinese Medicine, Nanjing 210023, China; zhangmmars@163.com (M.Z.); sen_twinklestar@126.com (J.Y.); fairyjsj@163.com (S.J.); 3Nanjing Research Center for Infectious Diseases of Integrated Traditional Chinese and Western Medicine, Nanjing 210001, China; 4College of Traditional Chinese Medicine·College of Integrated Traditional Chinese and Western Medicine, Nanjing University of Chinese Medicine, Nanjing 210023, China; kfweee@163.com; 5Department of Infectious Diseases, National Medical Center for Infectious Diseases and Shanghai Key Laboratory of Infectious Diseases and Biosafety Emergency Response, Huashan Hospital, Fudan University, Shanghai 200437, China; zhangwenhong@fudan.edu.cn

**Keywords:** SARS-CoV-2, breakthrough infection, reinfection, memory B cells, IFN-γ

## Abstract

The rapid mutation of SARS-CoV-2 has led to multiple rounds of large-scale breakthrough infection and reinfection worldwide. However, the dynamic changes of humoral and cellular immunity responses to several subvariants after infection remain unclear. In our study, a 6-month longitudinal immune response evaluation was conducted on 118 sera and 50 PBMC samples from 49 healthy individuals who experienced BA.5/BF.7/XBB breakthrough infection or BA.5/BF.7-XBB reinfection. By studying antibody response, memory B cell, and IFN-γ secreting CD4^+^/CD8^+^ T cell response to several SARS-CoV-2 variants, we observed that each component of immune response exhibited distinct kinetics. Either BA.5/BF.7/XBB breakthrough infection or BA.5/BF.7-XBB reinfection induces relatively high level of binding and neutralizing antibody titers against Omicron subvariants at an early time point, which rapidly decreases over time. Most of the individuals at 6 months post-breakthrough infection completely lost their neutralizing activities against BQ.1.1, CH.1.1, BA.2.86, JN.1 and XBB subvariants. Individuals with BA.5/BF.7-XBB reinfection exhibit immune imprinting shifting and recall pre-existing BA.5/BF.7 neutralization antibodies. In the BA.5 breakthrough infection group, the frequency of BA.5 and XBB.1.16-RBD specific memory B cells, resting memory B cells, and intermediate memory B cells gradually increased over time. On the other hand, the frequency of IFN-γ secreting CD4^+^/CD8^+^ T cells induced by WT/BA.5/XBB.1.16 spike trimer remains stable over time. Overall, our research indicates that individuals with breakthrough infection have rapidly declining antibody levels but have a relatively stable cellular immunity that can provide some degree of protection from future exposure to new antigens.

## 1. Introduction

According to data from the World Health Organization (WHO), since the outbreak of COVID-19 in December 2019, the world has been working hard to cope with a serious health crisis that has affected over two billion people worldwide and claimed over 6.8 million lives [1]. Although the WHO has declared that SARS-CoV-2 no longer constitutes a public health emergency of international concern, Omicron with its subvariants has caused many waves of infection worldwide [2]. The spike proteins of BA.5 and BF.7 virus have only one amino acid difference, which became predominant worldwide by the middle of 2022 [3]. The XBB clade has rapidly evolved into several sub-lineages, including XBB.1.5 and XBB.1.16, which became predominant worldwide by the middle of 2023 [4]. These three variants were responsible for a surge of infections in many countries [5,6]. The duration of herd immunity established after BA.5/BF.7/XBB infection or BA.5/BF.7-XBB reinfection, and its protection against newly emerging variants, remain uncertain. Due to the rapid evolution of SARS-CoV-2, Omicron variants have continued to give rise to many sub-lineages (for example EG.5.1, HK.3, BA.2.86 and JN.1) [7,8]. More evidence has suggested that significant changes in their antigen characteristics enabled these variants to evade serum neutralization in vaccinated and convalescence individuals [9,10]. Therefore, the evolving variant waves still have a great impact on public health.

As is well known, the humoral and cellular immune responses elicited by SARS-CoV-2 infection can lead to virus clearance and prevent serious diseases [11]. High antibody titers are associated with the prevention of breakthrough infection by constantly mutating variants, but do not provide comprehensive immunity [12,13]. The potent MBC response typically contributes to the production of high levels of neutralizing antibodies, which can prevent reinfection by newly emerging variants [14,15,16]. T cell response is crucial for achieving virus clearance and limiting disease severity [17,18]. With the complete liberalization of epidemic prevention measures for COVID-19, numerous individuals are being infected by various strains within the same year [19,20]. The antigenic differences between variants indicate that different variant infections might generate differential immune memory [21,22].

Our previous research has confirmed that individuals with breakthrough infections of BA.5/BF.7/XBB have relatively high neutralizing antibody responses [23]. Given that the dynamics, duration, and evolution of immune memory during infections are often unpredictable, and the short-term immune response after the resolution of infection has low predictive value for long-term protection [24,25], there is an urgent need to simultaneously evaluate the antibody, memory B cell, and CD4^+^/CD8^+^ T cell dynamics responses to distinct SARS-CoV-2 variants for individuals with hybrid immunity.

## 2. Materials and Methods

### 2.1. Serum Samples

For all COVID-19 participants, the clinical diagnosis criteria were based on the 9th National COVID-19 guidelines. Among the participants, 13 had BA.5 infection, 10 had BF.7 infection and 13 were infected with the XBB variant. These 13, who had previously been reinfected with XBB virus following BA.5/BF.7 breakthrough infection after receiving three doses of inactivated vaccine (BBIBP-CorV or CoronaVac), were recruited at the Nanjing Hospital of Chinese Medicine, Nanjing, China. The SARS-CoV-2 infection of all the subjects was confirmed by Polymerase Chain Reaction (PCR). All the volunteers had a “mild” case of COVID-19, with mild clinical symptoms and no signs of pneumonia on imaging, not requiring hospitalization. Their baseline characteristics are summarized in Table 1. All the participants provided written informed consent. All activity was conducted according to the guidelines of the Declaration of Helsinki and approved by the ethical committee of Nanjing Hospital of Chinese Medicine Affiliated to Nanjing University of Chinese Medicine (number KY2021162). Six volunteers (two with BA.5 infection, three people with XBB infection, and one with BA.5/BF.7-XBB reinfection) did not provide samples 6 months after infection.

### 2.2. Cell Lines

Expi293F cells (Thermo Fisher, Shanghai, China, Cat# A14527) were cultured in the serum free SMM 293-TI medium (Sino Biological Inc., Beijing, China) at 37 °C with 8% CO_2_ on an orbital shaker platform. HEK293T cells (Cat# CRL-3216) and Vero E6 cells (cat# CRL-1586) were from ATCC and cultured in 10% Fetal Bovine Serum (FBS, GIBCO cat# 16140071), supplemented with Dulbecco’s Modified Eagle Medium (DMEM, ATCC cat# 30-2002) at 37 °C, 5% CO_2_. In addition, I1 mouse hybridoma cells (ATCC, cat# CRL-2700) were cultured in Eagle’s Minimum Essential Medium (EMEM, ATCC cat# 30-2003) with 20% FBS.

### 2.3. Construction and Production of Variant Pseudoviruses

Omicron sub-lineage spikes and the WT (D614G) SARS-CoV-2 spike were constructed into plasmids. The amino acid sequence of the full-length spike protein of SARS-CoV-2 WT and the mutation site of the Omicron subvariants are listed in Table 2. These identified spike plasmids were transfected into HEK293T cells using PEI (Polyscience, Shanghai, China). The cells were cultivated overnight at 37 °C with 5% CO_2_. Then, the cells were infected with VSV-G pseudo-typed ΔG-luciferase (G*ΔG-luciferase, Kerafast, Shanghai, China) at MOI = 5. After four hours, cells were washed three times with 1 × DPBS. The transfection supernatant was collected and centrifuged at 3000× *g* for 10 min the following day. Then, I1 hybridoma (anti-VSV-G; ATCC, CRL-2700) supernatant was added into every viral stock and incubated for one hour at 37 °C in order to neutralizing excess VSV-G. Titer measurements were then performed, and aliquots were made to be stored at −80 °C.

### 2.4. Pseudovirus Neutralization Assays

In order to perform neutralization experiments, pseudoviruses were incubated with serial dilutions of sera, and the reduction in luciferase gene expression was determined. Briefly, 2 × 10^4^ Vero E6 cells were seeded per well in a 96-well plate. On the next day, the pseudoviruses were incubated with triple serial dilutions of the test samples for 30 min at 37 °C. Then, the cells were added into the mixture and cultivated for another 24 h. The Luciferase Assay System (Beyotime, Shanghai, China) was used to measure the luminescence. The half-maximum concentration (IC_50_) was defined as the dilution for which the relative light units were reduced by 50% compared with the virus control (virus + cells) after subtracting the background of the blank control groups (cells only). The IC_50_ values were calculated using nonlinear regression in GraphPad Prism. The half-maximal inhibitory concentration (IC_50_) represents the dilution factor that reduces the relative light unit by 50% compared to the virus control (virus + cell) after subtracting the background of the blank control group (only cells). To calculate this parameter, we used the scientific computing software GraphPad Prism v.10 to calculate the IC_50_ value through nonlinear regression.

### 2.5. Protein Expression and Purification

The WT/BA.5/XBB.1.16 RBD (aa319-541) and the SARS-CoV-2 S2P S trimer proteins were cloned independently into the mammalian expression vector pCMV3, incorporating an Avi-tag and an 8 × His tag at the C terminus. S trimer or RBD expressing plasmids were transfected into Expi293F cells using PEI to express proteins. The cells were then cultured at 37 °C, shaking at 125 rpm with 8% CO_2_ for 5 days. The supernatant was collected after centrifuging at 4000× *g* for 10 min and Ni-Smart beads (Smart-Lifesciences, SA035050, Shanghai, China) were added and incubated for 1 h at room temperature. Ten column volumes of 20 mM imidazole in PBS were used to wash the beads, and 500 mM imidazole in PBS was used for elution. The protein was purified, buffer swapped into PBS without imidazole, concentrated using a 10 kDa cutoff column, and its purity was evaluated using SDS-PAGE and OD_280_. 1 mg/mL Avi-labeled RBD was incubated with 10 mM ATP, 10 mM magnesium acetate, 50 μM D-biotin, 50 mM Bicine, and 19.2 μg/mL purified BirA at 30 °C for five hours for enzymatic biotinylation.

### 2.6. Memory B Cells Detection by Flow Cytometry

We used a PBMC isolation kit (Solarbio, P8900, Shanghai, China) to isolate PBMCs. First, in order to detect antigen-specific SARS-CoV-2 B cells, approximately 1.0 × 10^6^ PBMCs were treated with LIVE/Dead Zombie Aqua Dye (423102, BioLegend; Beijing, China, 1 μL/well) at 4 °C for 15 min. Biotinylated SARS-CoV-2 RBD protein was combined with Streptavidin-BV421 (405225, BioLegend) at a 4:1 molar ratio and let to sit for one hour at 4 °C to obtain the RBD probe. Fluorescence-Activated Cell Sorting (FACS) buffer (PBS + 2% FBS) was used to resuscitate PBMCs into a cell suspension, then the sample was transferred to a microplate, ensuring 100 μL per well. Then, at 4 °C, antigen probes with a concentration of 1.5 μL per well were used for staining for 30 min, followed by these conjugated antibodies: APC anti-human CD3 (300412, BioLegend; 1 μL/well), PerCP/Cyanine 5.5 anti-human CD19 (302230, BioLegend; 1 μL/well), FITC anti-human CD21 (354910, BioLegend; 1 μL/well), PE anti-human CD27 (302808, BioLegend; 1 μL/well). After being stained, the cells were washed and resuspended in 200 μL FACS buffer. The full gating strategy is illustrated in Appendix A.

### 2.7. T Cells Detection by Flow Cytometry

In order to detect SARS-CoV-2-specific T cells, 2.0 × 10^6^ PBMCs were prepared in a cell suspension, added to plates (250 μL/well), and then incubated with CD28 Monoclonal Antibody (CD28.2) (302934, BioLegend; 0.5 μL/well), CD49d (Integrin alpha 4) Monoclonal Antibody (9F10), (304340, BioLegend; 0.25 μL/well), Monensin (420701, BioLegend; 0.25 μL/well) and 500 μg/mL SARS-CoV-2 S trimer (2 μL/well) overnight at 37 °C with 5% CO_2_, after being cleaned and resuspended with the RPMI-1640 (Giboco, 11875093, Shanghai, China) containing 10% FBS. The assay was validated by incubating an equivalent quantity of PBMCs with either PBS as the negative control or PMA + Ionomycin (423302, BioLegend; 0.5 μL/well) as the positive control. The cells were made into a cell suspension and transferred to plates (50 μL/well) after washing and resuspending with FACS buffer. PBMCs were incubated with LIVE/Dead Zombie Aqua Dye (423102, BioLegend; 1 μL/well) for 15 min at 4 °C, and then stained for 30 min at 4 °C using the following antibodies: Brilliant Violet 711™ anti-human CD4 (344648, BioLegend; 1 μL/well), APC anti-human CD3 (300412, BioLegend; 1 μL/well), and PE/Dazzle™ 594 anti-human CD8 (301058, BioLegend; 1 μL/well). After washing with FACS buffer, the cells were stained for 20 min at 4 °C using a Cyto-Fast™ Fix/Perm Buffer Set (426803, BioLegend; 100 μL/tube). After being washed and resuspended with Cyto-Fast™ perm wash solution, cells were prepared into a cell suspension, added to plates (50 μL/well), and then stained for 20 min at 4 °C using the following antibody: Brilliant Violet 605™ anti-human interferon-gamma (IFN-γ) (506542, Biolegend; 1 μL/well). After being stained, the cells were washed and resuspended in 200 μL FACS buffer. FlowJo software v10.10 was used for the analysis of the B and T cell populations. The full T cells gating strategy is illustrated in Appendix A.

### 2.8. ELISA

We coated a 100 ng per well of S trimer or 100 ng per well of RBD onto ELISA plates at 4 °C overnight. The ELISA plates were then blocked with 300 μL blocking buffer (3% BSA in PBS) at 37 °C for two hours. Afterwards, sera were serially diluted using dilution buffer (3% BSA in PBS), incubated at 37 °C for one hour. Next, 100 μL of 2000-fold diluted goat anti-human IgG (H + L) antibody (Beyotime, Shanghai, China) was added into each well and incubated for one hour at 37 °C. The plates were washed between each step with PBST (0.5% Tween-20 in PBS). Finally, the TMB substrate (Beyotime, Shanghai, China) was added and incubated before the reaction was stopped using 1 M sulfuric acid. Absorbance was measured at 450 nm. OD_450_ readings were normalized by subtracting the average of negative control wells and finally dividing by the average maximum signal for each unique coating protein in each experiment. Normalized OD_450_ data from biological replicates was combined and fitted with a three-parameter logistic model.

### 2.9. Quantification and Statistical Analysis

GraphPad Prism v.10 (GraphPadSoftware, Santiago, MN, USA) was utilized for statistical analyses of the pseudovirus neutralization and antibody binding evaluations, in order to determine the average value for every data point. Every specimen underwent analysis three times. Antibody neutralization IC_50_ and binding EC_50_ values were determined using a five-parameter dose–response curve in GraphPad Prism. All the fold changes are calculated based on the comparison of geometric mean titers. For comparing serum neutralization titers, RBD^+^MBCs, IFN-γ^+^CD8^+^T cells, IFN-γ^+^CD4^+^T cells and binding titers, statistical analysis was performed using Multiple Mann–Whitney tests. *p* values with two tails are provided. To ascertain whether the data satisfied the statistical approach’s presumptions, no statistical techniques were applied.

## 3. Results

### 3.1. COVID-19 Breakthrough Infection Cohorts

To investigate the long-term changes in immune response to circulating variants, we recruited four different groups consisting of 49 volunteers from Nanjing Hospital of Chinese Medicine (Figure 1). All the volunteers had a “mild” case of COVID-19, not requiring hospitalization. Of these, 13 volunteers were infected with BA.5 virus following three doses of inactivated vaccine. The sera and PBMC samples were collected at different time points (14 days, 2 months, 4 months, and 6 months) post BA.5 breakthrough infection, 10 volunteers received three doses of the inactivated vaccination before infecting the BF.7 virus, and sera samples were collected at Day 14 and 4-month post BF.7 breakthrough infection. Then, 13 volunteers were infected with XBB virus following three doses of inactivated vaccine, and sera samples were collected at Day 14 and 6-month post XBB breakthrough infection. At the same time, 13 volunteers were reinfected with XBB virus following BA.5/BF.7 breakthrough infection, and sera samples were collected at Day 14 and 6-month post XBB reinfection. Baseline characteristics of the participants are listed in Table 1.

### 3.2. Anti-Spike and RBD Specific Antibody Magnitude and Durability Elicited by Breakthrough Infection

In order to evaluate the antibody response to breakthrough infection, 118 sera samples were collected from 49 volunteers according to the study design (Figure 1). To measure the binding antibody levels for individuals infected with different variants, we constructed three SARS-CoV-2 Spike trimers (WT/BA.5/XBB.1.16) and RBD. All the sera samples were firstly tested for the presence of anti-Spike and anti-RBD antibodies by ELISA. We found that the levels of anti-WT/BA.5/XBB.1.16 Spike and RBD IgG binding titers all remained at the maximum level at Day 14 after BA.5 breakthrough infection (Figure 2A and Appendix A). A significant decrease in anti-WT/BA.5/XBB.1.16 Spike and RBD IgG antibody levels was observed at Day 60, and the same for Day 120. A more significant decline was observed by Day 180. However, prominent anti-SARS-CoV-2 WT/BA.5/XBB.1.16 spike or RBD IgG responses were still detectable in all individuals who were followed (6 months) after the virus breakthrough infection. As shown in Figure 2B and Appendix A, the geometric mean titers (GMTs) of anti-Spike/RBD binding antibodies from the BA.5 infection group showed a continuously downward trend over time against all tested variants, with WT showing the fastest decrease (Spike: slope = 16.04; RBD: slope = 8.11).

We compared the fold change of anti-SARS-CoV-2 WT/BA.5/XBB.1.16 spike or RBD IgG levels between Day 14 and Day 120 or 180 across different variants infection groups. The geometric mean fold change of anti-WT/BA.5/XBB.1.16 spike IgG levels displayed a relatively similar trend between the four groups (Figure 2C), while the fold change of anti-WT/BA.5/XBB.1.16-RBD IgG levels from the BA.5 breakthrough infection group was the highest among all groups (Appendix A).

### 3.3. Neutralizing Antibody Titers against SARS-CoV-2 Variants

According to our previous research, breakthrough infections have been confirmed to significantly increase the neutralizing antibody titer against WT and Omicron subvariants [23,26,27]. However, there is still a lack of longitudinal breakthrough infection serological data for emerging Omicron subvariants, especially for newly evolved variants. In this study, we constructed pseudoviruses (PsVs) of prevalent Omicron sub-lineages (including EG.5.1, HK.3, BA.2.86, and JN.1). First, we conducted a longitudinal serum sample evaluation for up to 6 months in individuals infected with BA.5/BF.7/XBB or reinfected with BA.5/BF.7-XBB. This evaluation mainly focused on the neutralizing activity of WT and Omicron subvariants. During the research on the breakthrough infection BA.5, we observed that the neutralizing antibody titer against each variant reached a relatively high level on day 14. However, as time went on, the level of these neutralizing antibodies significantly decreased from their peak. This means that, with the passage of time, antibody resistance may gradually weaken, thus increasing the risk of reinfection. However, for early strains (including WT, BA.2, BA.5, and BF.7), most individuals can maintain a certain neutralizing titer. However, when we turn our attention to the six months after breakthrough infection, they almost completely lose their resistance to the neutralizing activity of several subvariants of Omicron BQ.1.1, CH.1.1, BA.2.86, JN.1, and XBB (Figure 3A).

To explore whether there is a difference in the decline rate of neutralization GMTs against different variants over time, we calculated the decrease slope in neutralization GMTs for different variants within the BA.5 breakthrough infection group. We found that the neutralization GMTs exhibited a continuous decreasing trend from Day 14 to 180 among all tested variants, with WT showing the fastest decline rate (Figure 3B).

We then compared the neutralization GMTs against each variant at Day 14 after different variants infection. Similar to our previous study [23,26], Omicron BQ.1.1, CH.1.1, and XBB.1.5 comparably evaded breakthrough infection sera at Day 14. Two sub-lineages that evolved from XBB, EG.5.1 showed a similar level of evasion to that of XBB.1.5, while HK.3 showed even lower neutralization titers than those of XBB.1.5 on Day 14. The JN.1 variant did not demonstrate greater resistance to neutralization than BA.2.86. Sera from BA.5/BF.7-XBB reinfection displayed a shifted neutralization pattern, showing the highest neutralizing GMT of 3908 against BA.5. BA.5/BF.7-XBB reinfection also induced a significantly higher level of neutralizing antibodies against XBB and BA.2.86 subvariants compared with other three groups. These results indicated that neutralizing antibody titer was related to the antigenic component of prior exposure (Figure 3C).

To investigate whether there is a difference in the decrease rate of neutralizing antibody titers among four groups, we compared the geometric mean fold change of serum neutralization ID_50_ at Day 14 versus 4 or 6 months for all tested PsVs. The geometric mean fold change of serum neutralization ID_50_ from BA.5 and BA.5/BF.7-XBB infection groups exhibited relatively higher levels, while the XBB infection group showed the lowest fold change (Figure 3D).

Additionally, we investigated the relationship between binding and neutralizing antibody titers for every group on Day 14. Spike- and RBD-binding antibody titers against the WT and BA.5 viruses showed a consistent connection, but not for the XBB.1.16 virus (Appendix A). When comparing Spike/RBD binding and neutralization antibody titers against each virus, hardly any discernible association was found (Appendix A). These findings indicate that it is difficult to predict a certain correlation between Spike/RBD binding and neutralizing antibody responses due to the evolution of the virus and the increased complexity of hybrid immunity.

### 3.4. Dynamic Changes of RBD-Specific Memory B Cells to Breakthrough Infection

After evaluating the dynamic evolution of antibody response to breakthrough infections of BA.5, BF.7, XBB, and reinfection with BA.5/BF.7-XBB, we further focused on comparing the dynamic changes in cellular immune response at multiple time points in 50 PBMC samples selected from the BA.5 convalescents. By using flow cytometry, we measured the number of SARS-CoV-2-specific peripheral blood B cells and T cells in these groups (Appendix A). In the process of exploring the detection of SARS-CoV-2 specific memory B cells (MBCs), we used fluorescently labeled multimeric probes to identify RBD-specific B cells. We found that the frequency of RBD-specific MBCs conjugated to BA.5 and XBB.1.16, showed a significant increase in the B cell response to WT, BA.5, and XBB.1.16 viruses in BA.5 convalescents. On the contrary, the frequency of WT RBD specific MBCs decreased slightly over 6 months and tended to stabilize overall (Figure 4A). However, comparing MBCs across WT, BA.5, and XBB.1.16 over all time points (Appendix A), it appears that, at day 14, MBC detection of the more recent variations is reduced, especially for XBB.1.16. Overall, based on the observations here, the development of memory B cells targeting SARS-CoV-2 induced by virus breakthrough infection was robust, and is likely to be long lasting. We further divided CD19^+^ B cells into four subgroups, and the frequency of resting (CD27^+^CD21^+^) and intermediate (CD27^−^CD21^+^) MBCs increased over the 6 months. Conversely, the frequency of activated (CD27^+^CD21^−^) and atypical (CD27^−^CD21^−^) MBCs decreased over the 6 months (Figure 4B–D). These results indicate that MBC responses are gradually mature and stable as time goes by. Considering that MBCs could rapidly transform into antibody secreting cells upon encountering antigens again, breakthrough infections may awaken individual persistent immune memory function.

### 3.5. Dynamic Changes of Spike-Specific CD4^+^ and CD8^+^ T Cell Response Elicited by Breakthrough Infection

Preventing severe symptoms and controlling viral infections necessitates an understanding of how SARS-CoV-2-specific T-cell responses in convalescent individuals change dynamically [18,28]. SARS-CoV-2 WT, BA.5, and XBB.1.16 S trimers were used as stimulations to measure different patterns in S-specific IFN-γ secretion among all BA.5 convalescents at different time points. There were no significant changes in the proportions of S-specific IFN-γ-producing CD4^+^ T cells up to 6 months after infection, as shown in Figure 5A. Similarly, the frequency of IFN-γ-producing CD8^+^ T cells did not significantly differ up to 6 months post-infection (Figure 5B). The percentages of IFN-γ^+^CD4^+^ or IFN-γ^+^CD8^+^ did not differ significantly among WT, BA.5, and XBB.1.16 stimulations (Appendix A), which suggests that even the most recent variants can still be detected by T cells. In summary, our data indicates that CD4^+^ and CD8^+^ T cell responses are relatively stable for at least 6 months, which can provide sufficient protection against future infections and severe illness occurrence.

## 4. Discussion

Although more than four years have passed since the COVID-19 pandemic, the continuous evolution and spread of the new SARS-CoV-2 variants still pose a major threat to the health of people around the world [29,30]. By reason of the strong immune escape ability of Omicron and its variants, many cases of breakthrough infections have occurred in populations undergoing rehabilitation and vaccination [31,32]. Yet there are still some questions, including differences in antibody response between variant infections and waning of protection against infection over a period of several months.

In this study, we described the dynamic changes of humoral and cellular immune responses in volunteers with diverse hybrid immunity histories. To assess the dynamic change of serum-neutralizing activities, we assembled a total of 12 variants, including both those that were widely disseminated in the past and the currently circulating XBB sub-lineages, as well as the most recently identified JN.1 [33]. According to our knowledge, this panel of SARS-CoV-2 variants for longitudinal analysis is thought to be the most comprehensive. We found that the levels of IgG and neutralizing antibodies decreased from peak levels but were still detectable in most subjects over the next 4–6 months. Consistent with previous research findings [34,35,36], antibodies obtained from breakthrough infections can persist for more than 6 months against early Omicron variants. Although our research did not involve the changes in antibody levels of infected people who have not been vaccinated, as almost everyone has been immunized through vaccination or infection, we can infer that the antibody level and persistence of these people with hybrid immunity may not significantly differ from those of infected individuals without vaccination [37,38]. There are several research examples showing that, compared with people who have been infected with SARS-CoV-2 but have not been vaccinated, hybrid immunity can significantly improve the probability of preventing severe cases and hospitalization [39,40,41]. Therefore, before facing the next infection, booster vaccination can provide additional protection for people who have been infected with SARS-CoV-2. More importantly, post-Omicron variants, such as BQ, CH, XBB, BA.2.86, and JN.1 variants, exhibit severe immune evasion, with neutralizing antibody levels reaching undetectable levels within 4–6 months. These results indicate that, because of the rapid mutation of SARS-CoV-2, it is difficult for previous vaccinations and breakthrough infections to prevent reinfection after several months. We also found that additional XBB reinfection significantly improved the levels of neutralizing antibodies against Omicron subvariants. This may be related to somatic hypermutation and antibody affinity maturation of memory B cells that drift towards the previously contacted antigens [42,43]; once encountering antigen stimulation again, high-level antibodies targeting the relevant variants can be rapidly produced [44]. Our findings support that the influence of immune imprinting can be eliminated by multiple exposure to different antigens [21], and further emphasizes the necessity of taking up effective vaccination strategies in previously infected individuals to enhance immune responses against emerging variants and to combat the ongoing COVID-19 pandemic [7,45].

Our research also focuses on the lifespan of specific antiviral antibody reactions. The development of memory B and T cells is crucial for long-term protection, but the longitudinal dynamics of cellular immunity targeting to several variants remain unresolved [46,47]. In a positive sense, although serum IgG antibody levels decreased over time in our study, specific memory B cells maintained and further increased over the 6 months period. Importantly, we found that S-specific IFN-γ producing cells are present in all BA.5 breakthrough infection samples. Furthermore, we showed that S-specific T cell responses were not only detectable but also appeared to be maintained over 6 months. These findings highlight the persistence of virus-specific cellular immunity. Therefore, a better evaluation of the effectiveness and longevity of the immune response against the virus can be achieved by studying these responses over a longer period.

There are still several limitations in our research. Firstly, we recruited a relatively small number of volunteers; all the collected longitudinal samples are not the same, although statistical analysis was conducted, and this may have an impact on the results. Secondly, we did not conduct experimental validation on deep specific B and T cell immune responses. PBMCs were only collected from BA.5-infected individuals. Thirdly, our study did not collect samples of volunteers before infection, so the immune response of these volunteers to the inactivated vaccine before infection was not clear. Finally, the detection assays conducted in our study used pseudoviruses rather than authentic viruses.

## 5. Conclusions

In summary, our study demonstrates that emerging Omicron sub-lineages show significant neutralization escape and the serum antibody responses of individuals with different immunological backgrounds gradually decrease over time. Encouragingly, there are high levels of long-lived memory T and B cells in the body that can be reactivated after the second exposure, providing sufficient immune protection [45]. Thus, this highlights again that, with the presence of strong immune memory, booster vaccination can provide essential population protection as we continue to combat the rapidly evolving virus.

## Figures and Tables

**Figure 1 vaccines-12-00464-f001:**
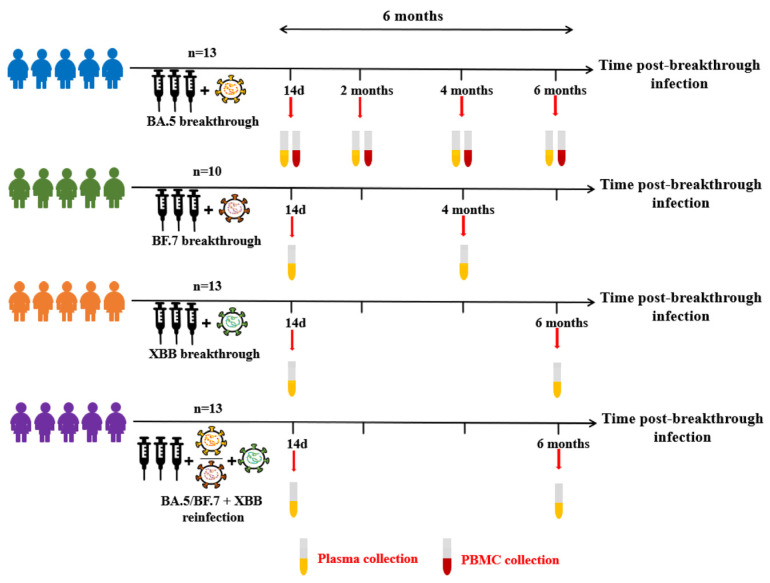
Study Design and Sample Collection.

**Figure 2 vaccines-12-00464-f002:**
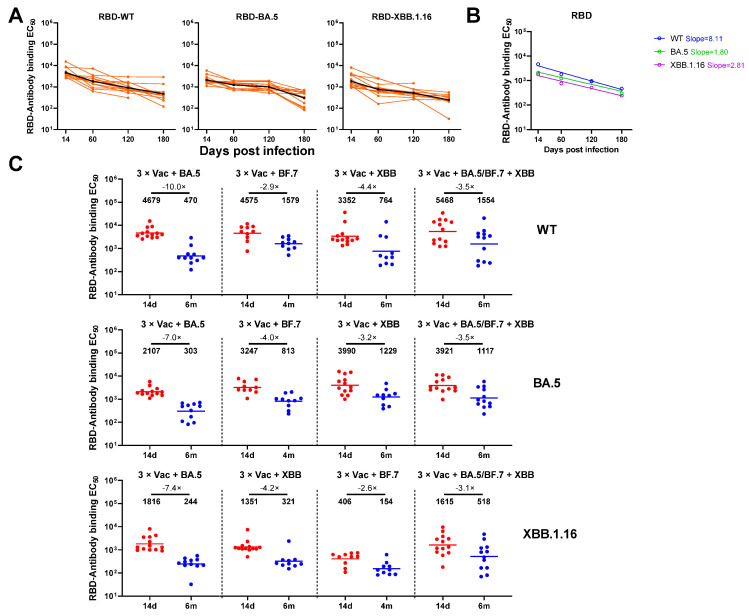
Anti-spike binding antibody responses after BA.5/BF.7/XBB breakthrough infection or BA.5/BF.7-XBB reinfection following three doses of vaccination. (**A**) Serum anti-WT/BA.5/XBB.1.16 Spike IgG levels over time for individuals with BA.5 breakthrough infection. (**B**) Decline rate of binding GMTs by sera collected from individuals with BA.5 infection against WT/BA.5/XBB.1.16 spike over time. (**C**) Parallel comparison of anti-WT/BA.5/XBB.1.16 Spike IgG levels at Day 14 and 4 or 6 months after BA.5/BF.7/XBB breakthrough infection or BA.5/BF.7-XBB reinfection. For all panels, values above the symbols denote geometric mean titer.

**Figure 3 vaccines-12-00464-f003:**
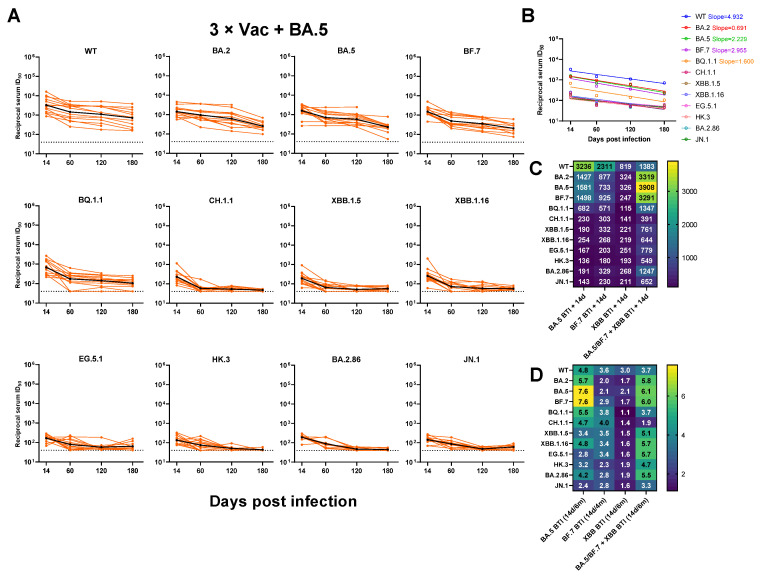
The dynamic changes of neutralizing antibody titers against SARS-CoV-2 variants from individuals after infection. (**A**) Serum neutralization ID_50_ over time for individuals after BA.5 breakthrough infection following three doses of inactivated vaccine against PsVs of WT, Omicron sub-lineages. (**B**) Decline rate of neutralization GMTs from individuals with BA.5 breakthrough infection against each PsV over time. (**C**) Heatmap for neutralization GMTs against each PsV by sera collected from individuals at Day 14 post BA.5/BF.7/XBB breakthrough infection and BA.5/BF.7-XBB reinfection. (**D**) Heatmap for geometric mean fold change of serum neutralization ID_50_ at Day 14 versus 4 or 6 months against different variants. The dashed line indicates the lower limit of detection of the neutralization assay, and values less than that were assigned as 40 for calculation. PsVs, pseudoviruses; WT, wildtype; BTI, breakthrough infection.

**Figure 4 vaccines-12-00464-f004:**
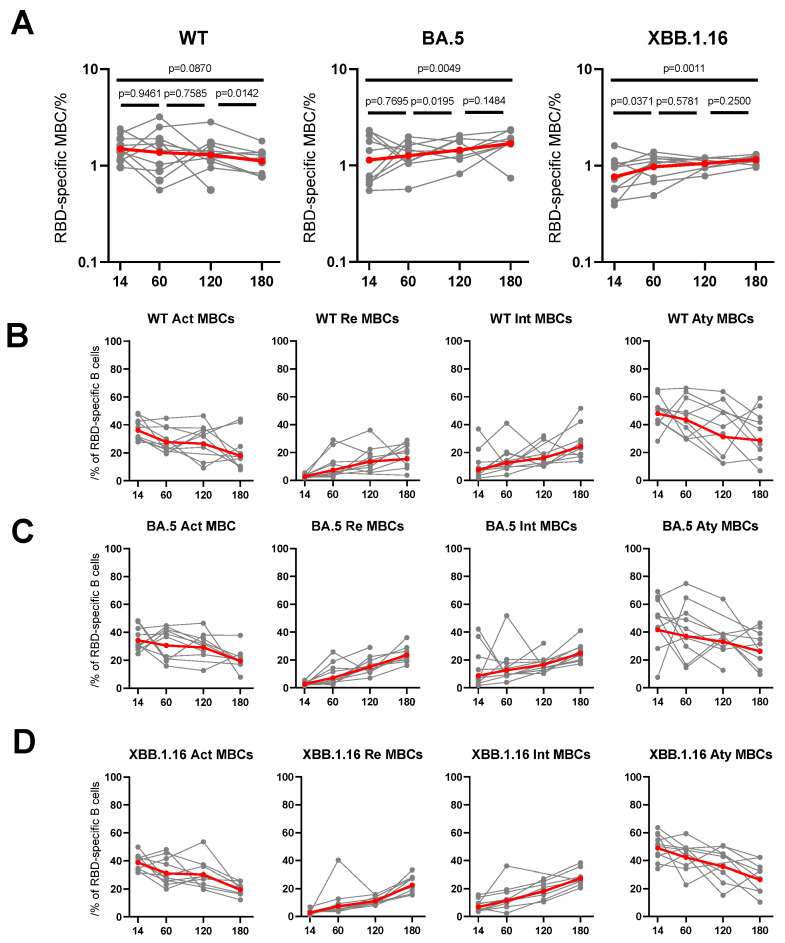
The dynamic changes of B cells’ responses to BA.5 breakthrough infection in individuals. (**A**) The frequencies of WT/BA.5/XBB.1.16 RBD^+^ MBCs at days 14, 60, 120, and 180 from individuals after BA.5 breakthrough infection. The frequencies of four subsets of WT (**B**), BA.5 (**C**), XBB.1.16 (**D**) RBD^+^ MBCs over time. For all, the numbers above plots represent the *p* values. The plots show mean values with 95% confidence interval. Statistics were calculated using Wilcoxon signed rank test for intragroup comparisons at different times. actMBCs, activate memory B cells; atyMBCs, atypical memory B cells; intMBCs, intermediate memory B cells; MBCs, memory B cells; rMBCs, resting memory B cells.

**Figure 5 vaccines-12-00464-f005:**
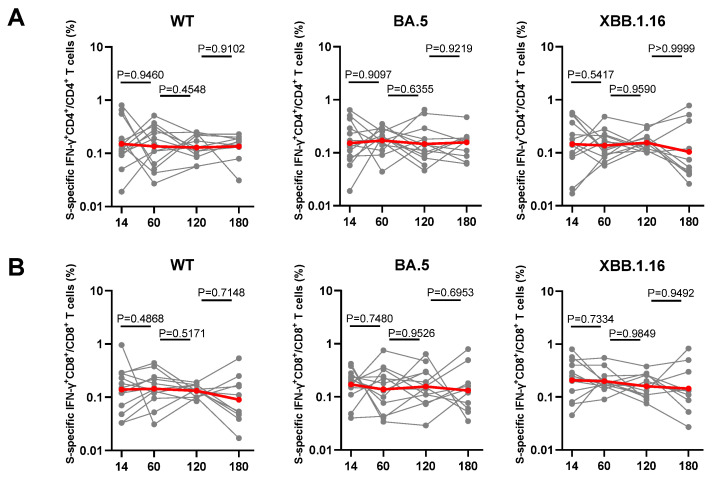
The dynamic changes of T cells response to BA.5 breakthrough infection in individuals. The frequencies of WT/BA.5/XBB.1.16 S-specific IFN-γ^+^ CD4^+^ (**A**) or IFN-γ^+^CD8^+^ T cells (**B**) at day 14, 60, 120, and 180 from individuals after BA.5 breakthrough (n = 13). In all instances, the numbers above plots represent the *p* values. The plots show mean values with 95% confidence interval. Statistics were calculated using Wilcoxon signed rank test for intragroup comparisons at different times. IFN-γ, interferon-gamma. grey line represent each individual, red line represent the average level of all individuals.

**Table 1 vaccines-12-00464-t001:** Baseline characteristics of enrolled participants.

	BA.5 Breakthrough Infection Individuals (n = 13)	BF.7 Breakthrough Infection Individuals (n = 10)	XBB Breakthrough Infection Individuals (n = 13)	BA.5/BF.7-XBB Reinfection Individuals (n = 13)
Age (year), median (range)	31.6 (22–47)	40.1 (34–50)	32.33 (19–46)	34.23 (18–69)
Male, n (%)	13 (59.09%)	4 (28.6%)	7 (53.84%)	6 (46.15%)
BMI (kg/m^2^), mean (SD)	23.0 (3.9)	23.9 (3.3)	23.3 (3.0)	21.4 (2.5)
Breakthrough infections months after the last COVID-19 vaccines, median (range)	10.8 (2–19)	10.6 (12–14)	16.6 (13–25)	10.6 (5–23)
Vaccinated days after the last COVID-19 vaccines, median (range)	427.2 (242–607)	407.4 (363–432)	535.8 (419–790)	337.7 (172–735)
Comorbidities (%)	N/A	N/A	N/A	N/A
Any, n (%)	0 (0%)	0 (0%)	1 (7.69%)	0 (0%)
HTN, n (%)	0 (0%)	0 (0%)	1 (7.69%)	2 (15.38%)
CAD, n (%)	0 (0%)	0 (0%)	0 (0%)	0 (0%)
DM, n (%)	0 (0%)	0 (0%)	0 (0%)	0 (0%)
NAFLD, n (%)	0 (0%)	0 (0%)	1 (7.69%)	0 (0%)
Hyperlipidemia, n (%)	0 (0%)	0 (0%)	0 (0%)	0 (0%)
Obesity, n (%)	2 (9.09%)	0 (0%)	0 (0%)	0 (0%)
Arrhy, n (%)	0 (0%)	0 (0%)	0 (0%)	0 (0%)
Asthma, n (%)	0 (0%)	0 (0%)	0 (0%)	0 (0%)
Rhinitis, n (%)	0 (0%)	0 (0%)	0 (0%)	0 (0%)
Urticaria, n (%)	0 (0%)	0 (0%)	0 (0%)	0 (0%)

BMI, body mass index. CAD, coronary artery disease. HTN, hypertension. DM, diabetes mellitus. Arrhy, arrhythmia, NAFLD, non-alcoholic fatty liver.

**Table 2 vaccines-12-00464-t002:** The amino acid sequence of the full-length spike protein of SARS-CoV-2 WT and the mutation site of the Omicron subvariants in this study.

Variant	Protein Sequence
WT	MFVFLVLLPLVSSQCVNLTTRTQLPPAYTNSFTRGVYYPDKVFRSSVLHSTQDLFLPFFSNVTWFHAIHVSGTNGTKRFDNPVLPFNDGVYFASTEKSNIIRGWIFGTTLDSKTQSLLIVNNATNVVIKVCEFQFCNDPFLGVYYHKNNKSWMESEFRVYSSANNCTFEYVSQPFLMDLEGKQGNFKNLREFVFKNIDGYFKIYSKHTPINLVRDLPQGFSALEPLVDLPIGINITRFQTLLALHRSYLTPGDSSSGWTAGAAAYYVGYLQPRTFLLKYNENGTITDAVDCALDPLSETKCTLKSFTVEKGIYQTSNF“RVQPTESIVRFPNITNLCPFGEVFNATRFASVYAWNRKRISNCVADYSVLYNSASFSTFKCYGVSPTKLNDLCFTNVYADSFVIRGDEVRQIAPGQTGKIADYNYKLPDDFTGCVIAWNSNNLDSKVGGNYNYLYRLFRKSNLKPFERDISTEIYQAGSTPCNGVEGFNCYFPLQSYGFQPTNGVGYQPYRVVVLSFELLHAPATVCGPKKSTNLVKNKCVNF”NFNGLTGTGVLTESNKKFLPFQQFGRDIADTTDAVRDPQTLEILDITPCSFGGVSVITPGTNTSNQVAVLYQGVNCTEVPVAIHADQLTPTWRVYSTGSNVFQTRAGCLIGAEHVNNSYECDIPIGAGICASYQTQTNSPRRARSVASQSIIAYTMSLGAENSVAYSNNSIAIPTNFTISVTTEILPVSMTKTSVDCTMYICGDSTECSNLLLQYGSFCTQLNRALTGIAVEQDKNTQEVFAQVKQIYKTPPIKDFGGFNFSQILPDPSKPSKRSFIEDLLFNKVTLADAGFIKQYGDCLGDIAARDLICAQKFNGLTVLPPLLTDEMIAQYTSALLAGTITSGWTFGAGAALQIPFAMQMAYRFNGIGVTQNVLYENQKLIANQFNSAIGKIQDSLSSTASALGKLQDVVNQNAQALNTLVKQLSSNFGAISSVLNDILSRLDKVEAEVQIDRLITGRLQSLQTYVTQQLIRAAEIRASANLAATKMSECVLGQSKRVDFCGKGYHLMSFPQSAPHGVVFLHVTYVPAQEKNFTTAPAICHDGKAHFPREGVFVSNGTHWFVTQRNFYEPQIITTDNTFVSGNCDVVIGIVNNTVYDPLQPELDSFKEELDKYFKNHTSPDVDLGDISGINASVVNIQKEIDRLNEVAKNLNESLIDLQELGKYEQYIKWPWYIWLGFIAGLIAIVMVTIMLCCMTSCCSCLKGCCSCGSCCK_
BA.2	WT+T19I, Del24-26, A27S, G142D, V213G, G339D, S371F, S373P, S375F, T376A, D405N, R408S, K417N, N440K, S477N, T478K, E484A, Q493R, Q498R, N501Y, Y505H, H655Y, N679K, P681H, N764K, D796Y, Q954H, N969K
BA.5	WT+T19I, Del24-26, A27S, Del69-70, G142D, V213G, G339D, S371F, S373P, S375F, T376A, D405N, R408S, K417N, N440K, L452R, S477N, T478K, E484A, F486V, Q498R, N501Y, Y505H, H655Y, N679K, P681H, N764K, D796Y, Q954H, N969K
BF.7	WT+T19I, Del24-26, A27S, Del69-70, G142D, V213G, G339D, R346T, S371F, S373P, S375F, T376A, D405N, R408S, K417N, N440K, L452R, S477N, T478K, E484A, F486V, Q498R, N501Y, Y505H, H655Y, N679K, P681H, N764K, D796Y, Q954H, N969K
BQ.1.1	WT+T19I, Del24-26, A27S, Del69-70, G142D, V213G, G339D, R346T, S371F, S373P, S375F, T376A, D405N, R408S, K417N, N440K, K444T, L452R, N460K, S477N, T478K, E484A, F486V, Q498R, N501Y, Y505H, H655Y, N679K, P681H, N764K, D796Y, Q954H, N969K
CH.1.1	WT+T19I, Del24-26, A27S, G142D, K147E, W152R, F157L, I210V, V213G, G257S, G339H, S371F, S373P, S375F, T376A, D405N, R408S, K417N, N440K, K444T, G446S, L452R, N460K, S477N, T478K, E484A, F486S, Q498R, N501Y, Y505H, H655Y, N679K, P681H, N764K, D796Y, Q954H, N969K
XBB.1.5	WT+T19I, Del24-26, A27S, V83A, G142D, Del144, H146Q, Q183E, V213E, G252V, G339H, R346T, L368I, S371F, S373P, S375F, T376A, D405N, R408S, K417N, N440K, V445P, G446S, N460K, S477N, T478K, E484A, F486P, F490S, Q498R, N501Y, Y505H, H655Y, N679K, P681H, N764K, D796Y, Q954H, N969K
XBB.1.16	WT+T19I, Del24-26, A27S, V83A, G142D, Del144, H146Q, E180V, Q183E, V213E, G252V, G339H, R346T, L368I, S371F, S373P, S375F, T376A, D405N, R408S, K417N, N440K, V445P, G446S, N460K, S477N, T478R, E484A, F486P, F490S, Q498R, N501Y, Y505H, H655Y, N679K, P681H, N764K, D796Y, Q954H, N969K
EG.5.1	WT+T19I, Del24-26, A27S, Q52H, V83A, G142D, Del144, H146Q, Q183E, V213E, G252V, G339H, R346T, L368I, S371F, S373P, S375F, T376A, D405N, R408S, K417N, N440K, V445P, G446S, F456L, N460K, S477N, T478K, E484A, F486P, F490S, Q498R, N501Y, Y505H, H655Y, N679K, P681H, N764K, D796Y, Q954H, N969K
HK.3	WT+T19I, Del24-26, A27S, Q52H, V83A, G142D, Del144, H146Q, Q183E, V213E, G252V, G339H, R346T, L368I, S371F, S373P, S375F, T376A, D405N, R408S, K417N, N440K, V445P, G446S, L455F, F456L, N460K, S477N, T478K, E484A, F486P, F490S, Q498R, N501Y, Y505H, H655Y, N679K, P681H, N764K, D796Y, Q954H, N969K
BA.2.86	WT+ Ins16MPLF, T19I, R21T, Del24-26, A27S, S50L, Del69-70, V127F, G142D, Del144, F157S, R158G, Del211, L212I, V213G, L216F, H245N, A264D, I332V, G339H, K356T, S371F, S373P, S375F, T376A, R403K, D405N, R408S, K417N, N440K, V445H, G446S, N450D, L452W, N460K, S477N, T478K, N481K, Del483, E484K, F486P, Q498R, N501Y, Y505H, E554K, A570V, H655Y, N679K, P681R, N764K, D796Y, S939F, Q954H, N969K, P1143L
JN.1	WT+Ins16MPLF, T19I, R21T, Del24-26, A27S, S50L, Del69-70, V127F, G142D, Del144, F157S, R158G, Del211, L212I, V213G, L216F, H245N, A264D, I332V, G339H, K356T, S371F, S373P, S375F, T376A, R403K, D405N, R408S, K417N, N440K, V445H, G446S, N450D, L452W, L455S, N460K, S477N, T478K, N481K, Del483, E484K, F486P, Q498R, N501Y, Y505H, E554K, A570V, H655Y, N679K, P681R, N764K, D796Y, S939F, Q954H, N969K, P1143L

The range of the RBD (319–541) is highlighted with double quotation marks. Ins: insertion, Del: deletion.

## Data Availability

Materials used in this study will be made available but may require execution of a materials transfer agreement. All the data are provided in the paper or the Appendix A.

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
