# Peer review of "Longitudinal Analysis of Humoral and Cellular Immune Response up to 6 Months after SARS-CoV-2 BA.5/BF.7/XBB Breakthrough Infection and BA.5/BF.7-XBB Reinfection"

_vaccines, 2024, doi:10.3390/vaccines12050464_

Round 1

Reviewer 1 Report

Comments and Suggestions for Authors

Since the pandemic, researchers have developed several vaccine platforms with diverse vaccination strategies. Despite initial protection against wild-type SARS-CoV-2 variants, the vaccine-induced response is not enough or marginal to protect against omicron variants. Publications on immune imprinting have shown that the type of vaccination influences immune responses. Wang et al., have submitted the study entitled “Longitudinal Analysis of Humoral and Cellular Immune Response up to 6 Months after SARS-CoV-2 BA.5/BF.7/XBB Breakthrough Infection and BA.5/BF.7-XBB Reinfection”. The authors collected kinetic samples (sera and PBMC) from 49 healthy individuals who received three doses of an inactivated vaccine and then experienced an Omicron variant infection, specifically BA.5, BF.7, and XBB 1.5. The results revealed that binding antibody titers have been reduced over time, whereas neutralizing antibody titers have not altered. Both T and B cells have maintained their antigen-specific cellular immunity within the same range over time.

 This reviewer has the following comments:

What is the inactivated vaccine? No information is provided.

Sections 2.3 and 2.5: Please provide the detailed sequences in the supplementary information, which are essential for reproducibility.

The authors have omitted the limitations of the study. For example, all the collected longitudinal samples are not the same. PBMCs were only collected from BA.5-infected individuals, not from the other groups. Only mild COVID-19 cases were considered. The immune response to inactivated vaccines before infection is not available. Use of pseudoviruses for the titration experiments.

Make sure you calculate the sample number correctly. For example, 4x13, 2x10, 2x13, 2x13 = 124. However, the authors say 132. Also, 78 PBMC samples, I can see 4x13 samples.

Figure 1: I am puzzled, IgG is produced by the plasma cells in vaccinated individuals. These antibodies should be available lifelong upon stimulation, unlike neutralizing antibodies. So why is there a drop against WT?

Figure 2: Statistics are missing.

Figure 3C and D panels are not legible; please enlarge them for clarity.

Section 3.4: B cell phenotyping should be designated. For example, resting memory B cells (CD19+???)….

Again, the B and T cell data show no significant differences in the percentages. If the antigen-specific B and T cells are not changing their specificity, why has the antibody (especially binding) titer been changing?

Author Response

Dear reviewer,

We thank you for the opportunity to revise and improve our manuscript. On behalf of my co-authors, we would like to express our great appreciation to you. We have complete all requires according to your suggestion. Note that changes made in the text are highlighted in red. Please see the attachment.

Kinds regards

Yours sincerely,

Yanliang Zhang (fsyy00404@njucm.edu.cn)

Department of Infectious Diseases, Nanjing Hospital of Chinese Medicine Affiliated to Nanjing University of Chinese Medicine, Nanjing, Jiangsu, China

Reviewer 2 Report

Comments and Suggestions for Authors

This manuscript delineates the humoral and cellular immune response with long-term follow-up in the Omicron subvariants breakthrough infection, including re-infection. The methodology was sound. The outcomes were satisfactory, and the results, presentation and visualisation were high quality. The content in this manuscript is novel and interesting to readers. However, the discussion was too short, and the argument on hybrid immunity vs convalescent without vaccination in different variants and subvariants was scant. The discussion should add more information to improve the quality of this manuscript.

Major concerns.

1. Lines 89-90 and Table 1: Suggest clarifying the COVID-19 vaccine used in the result (e.g. CoronaVac, BBIBP-CorV, ZF2001, BBV152…), not only vaccine types (e.g. inactivated platform). 

If all participants received the same vaccine regimen, there is no need to clarify Table 1.

2. How can we ensure the participants were the first infected with BA.5, BF.7, and XBB. breakthrough infection?
If they do not. The result from previous infections (e.g. WT, Delta...) and Omicron subvariants could broaden the immunity, altering the immunologic outcome, such as cross-reactivity.

Moreover, How do ensure in the BA.5/BF.7 + XBB infection?
Did some participants have previous infections before the Omicron?

Minor concerns.

1. Line 207: Suggest adding more "mild" classification definitions. Criteria for hospitalisation may differ depending on the policy and circumstances.

In some countries, all confirmed patients are hospitalised due to the "zero" or tightening of the measurement. In contrast to others, they admit only severe/life-threatening conditions regardless of pneumonia status due to "free" or limited resources.

I think at least additional information should clarify the pneumonia status of all participants in this study.

Comments.

1. Line 447: Suggest clarifying the statement by mimicking lines 95-96.

2. Line 90: Suggest adding the city and country of study site/facility to the statement.

3. Line 194: Suggest adding a version of the program, including the manufacturer name, city, and state.

4. Figure 2C: Was the statistical analysis based on the geometric ratio?

Suggest clarifying the Figure's description. Moreover, r, I suggest adding the statement to subsection 2.9 to clarify it because the subsection mentioned only Mann-Whitney.

5. Table 1: Suggest using "COVID-19 vaccines" instead of "coronavirus vaccines" to make it concise and clear.

Typos.

1. Lines 99, 102, 108 and 133: "COâ‚‚".

2. No spacing between temperature value and degree symbol. (e.g. "−80°C", not −80 °C)

3. Line 193: It seems to be an inconsistent format because the subsection was all capital letters without an abbreviation.

Author Response

(The authors gave the same response as above.)

Reviewer 3 Report

Comments and Suggestions for Authors

Dear Authors!

Thank you for the opportunity to review your manuscript "Longitudinal Analysis of Humoral and Cellular Immune Re- 2 sponse up to 6 Months after SARS-CoV-2 BA.5/BF.7/XBB 3 Breakthrough Infection and BA.5/BF.7-XBB Reinfection"

The COVID-19 infection, the immunization against it and study of immune response are very important due to high impact of this infection in whole population of the planet. Vaccination against COVID is an important tool for preventing sever and complicated cases and decrease the mortality rate.

The assessment of immune response is an essential for future vaccination programs and creating the personalized vaccination schedule

The aim of the study is cleare.

The Methods are described in details, each type of assessment organized as a single subparagraph. The methods are reproducible.

The results are complete, demonstrated with multiple figure and corresponded with study aims.

The discussion is relatively short for such amount of the findings.

I can recommend to make the discussion bigger

The manuscript required to organized the additional subsection " Limitations" at the end of the Discussion section

The Conclusion supports the findings of the study.

Author Response

(The authors gave the same response as above.)

Round 2

Reviewer 1 Report

Comments and Suggestions for Authors

The authors duly revised the manuscript by considering the reviewers' comments. Therefore, I now accept the manuscript for publication in its current form.